# The Degassing Processes for Oil Media in Acoustic Fields and Their Applications

**DOI:** 10.3390/polym14081497

**Published:** 2022-04-07

**Authors:** Alexander Karimov, Vladislav Bogdanov, Rim Valiullin, Ramil Sharafutdinov, Ayrat Ramazanov

**Affiliations:** 1Department of Electrophysical Facilities, National Research Nuclear University MEPhI, Kashirskoye hwy, 31, 115409 Moscow, Russia; bogdanov.starscream@mail.ru; 2Institute for High Temperatures, Russian Academy of Sciences, Izhorskaya 13/19, 127412 Moscow, Russia; 3Research Institute of Quality, Safety and Technologies of Specialized Food Products of the PRUE G.V. Plekhanov, Stremyanny Lane, 36, Building 6, 117997 Moscow, Russia; 4Department of Geophysics, Institute of Geology UFRC RAS, 16/2 Karl Marx Str., 450077 Ufa, Russia; valra@geotec.ru (R.V.); gframil@inbox.ru (R.S.); ramaz@geotec.ru (A.R.)

**Keywords:** colloidal system, dispersed bubbles, oil, heat of oil degassing, thermal fields

## Abstract

Numerous experiments on the effect of acoustic fields on oil media have shown the changing nature of oil physicochemical properties. In the present paper, we present a concept of internal airlift for oil medium with dissolved gas which could be propelled by external acoustic field. The mechanism determining gas bubble size as a function of pressure change is discussed. Model of interaction for the growing bubbles with acoustic fields is presented. Relationships specifying the characteristics of both the required acoustic field and oil medium are derived. The use of these relations makes it possible to define the available range of parameters for the system under consideration where one can obtain the expected effect on oil medium. It is demonstrated how the change in pressure and oil saturation (namely, the density of oil particles in the entire flow) of the medium is associated with temperature fields in the system. In particular, it is shown that the maximum deviation between the temperature change in oil and gas and gas–liquid media reaches a significant value, namely 10^−2^ K for a gas–liquid medium, while this difference is −0.1 K in an oil-and-gas medium. Using this approach, thermograms of oil producing wells have been analysed at a qualitative level.

## 1. Introduction

According to the modern classification [1,2,3], oil is classified as liquid dispersed system, with individual particles ranging from 10^−7^ (micro) to 10^−4^ cm (macro) in size. The dispersed phase of oil can be both solid and gaseous. Typically, the gas content of oil can vary from a few percent to 50%. Herewith, it should be kept in mind that physicochemical properties of an oil system in different dispersed states, even with the same composition, may differ significantly.

Why does it occur? One can point out a few reasons. As is known, the dispersed particles are an aggregate of a large number of particles bonded to each other by the relative weak intermolecular and interatomic interactions (the order of thermal energy) [3]. In a general case, they are in states far from thermodynamics equilibrium when the intermolecular bonds between colloidal particles can be weakened and strengthened. Such behavior should be determined by both the own processes of matter and energy transfer and some external factors [4]. Depending on these internal and external processes, the shape and size of such supra macromolecular structures should vary with time. Moreover, the nonstationary and nonlinear nature of these processes determines the complex character of the evolution of systems under consideration [5,6,7].

Such governing processes certainly include generation, growth and collapse of bubbles coming from the dissolved gas when the oil pressure changes as it moves up-wards along the well. Indeed, the pressure falls down from 100 atm at the reservoir to 1 atm at the wellbore [8,9]. In this case, the release of gas dissolved in oil can cause bubble growth. In fact, that is an internal airlift which may change the oil density and the heat transfer in this two-phase medium. This process may also propel the mechanically induced kinetic changes in supra macromolecular structures during the bubbles collapse, if any [10,11]. 

In this regard, the idea to enhance this process by acting on the oil medium with ultrasonic field arises. Although the effect of ultrasonic waves on oil has been long studied, this idea is only a conjecture. In order to have yet another attempt at proving this, we should consider the gas bubbles formation and evolution features in oil medium under the changing pressure and gas saturation. We should study the features of temperature field’s formation in such a two-phase medium, that is, we are going to determine the way of controlling the internal airlift using external acoustic fields in order to govern the macroscopic parameters of the entire volume. Proceeding from the results on the temperature distributions, we are also going to discuss the possibility of using this information for the diagnostics of oil fields.

## 2. Basic Physical and Chemical Properties of Oil Systems

Let us briefly dwell on the main physical and chemical properties of oil to estimate the expected structural changes [1,2,3]. As it is known, oil is a multicomponent, generally multiphase medium of low and high molecular weight species belonging to different homologous series. The low molecular weight compounds are mainly represented by paraffinic, naphthenic and aromatic hydrocarbons. The heavy part of the oil consists of high molecular weight paraffinic hydrocarbons and bicyclic aromatic hydrocarbons of the benzene and naphthalene series, resins and asphaltenes. It should also be kept in mind that, when the pressure in the medium decreases below the pressure of saturated vapors, they transform into a gaseous form. This process may naturally occur during the exploitation of oil deposits, owing to some processes in the deposit, and to oil lift. The exact composition of associated petroleum gas depends on the composition of the oil in which it is dissolved, as well as the conditions of its occurrence. However, as a rule of thumb, it contains hydrocarbons with an admixture of carbon dioxide or nitrogen, for which the degassing pressure lies in the range of 50–200 atm [12]. 

These all-oil components are characterized by different size, mass and dipole moments that define the variety of possible options for the establishment of intermolecular bonds due to the van der Waals forces, osmotic attraction and steric repulsion. For the illustration of such features, the typical particle size distribution in a liquid–liquid emulsion is shown in Figure 1. It is worth noting that the similar particle size distributions are seen in the oil medium [13,14,15].

The presence of particles of different sizes in the system leads to the fact that the medium acquires a random discrete structure consisting of space regions densely filled with particles and voids between them (see, for example, [13,17,18,19]). As an illustration of this structure, the typical micrograph of water-oil medium is shown in Figure 2 taken from [13].

In this regard it should be noted that with a hexagonal packing of a system consisting of spherical particles of the same size, the theoretically permissible degree of space filling is $g_{0}=72\%$ [17,18,19]. The value of g will depend primarily on the ratio of particles number with different sizes as well as on the intermolecular interactions which determine the filling of space with particles and the destruction/formation processes of particles. Depending on these parameters one can obtain the case of compact packing with $g>g_{0}$, or the case of incompact packing when $g<g_{0}$. Either way, the medium contains a significant number of voids.

Under certain conditions, these voids can change their configuration and size. So, when the pressure in the flow drops below a certain limit, they can be filled with gas dissolved in colloidal particles. Thus, gas bubbles could be generated in the medium. These bubbles are able to increase in scale, while the areas of local density rarefaction in the medium will expand until the moment when the surface energy of the bubble is sufficient to withstand pressure from other particles of the medium. 

It means that the oil medium can go into the state of a bubbly liquid flow, which already has properties that are different from the colloidal flow. In the case of a pressure gradient in the flow that is sufficient to destroy the bubbles, the surface energy accumulated in them, together with the heat released during their collapse, can clearly affect the structural composition of the medium. This is possible if the energy accumulated in a unit volume of the bubbly liquid is sufficient to destroy the chemical bond of the colloidal particles of the oil medium.

## 3. Quasi-Equilibrium Dynamics of Bubbles

Although the system of liquid–bubbles under consideration is in general case a thermodynamically nonequilibrium one, a certain equilibrium is set between each bubble and the surrounding liquid for short time. This gives us a rough model showing what may happen with the oil flowing from the formation up the well while its pressure drops from 100 to 10 atm. Here, the effect of oil degassing is neglected. Certainly, that is a very rough approach, but it shows what may happen with oil in general. In order to make such estimations we use the Laplace equation
(1)$$p_{l}=p_{g}-\frac{2\sigma }{R},$$
which describes the relationship between the gas pressure $p_{g}$ inside the bubble with radius R having the surface tension coefficient of the oil $\sigma =26\times 10^{-3}\;N/m$ [2] and the pressure $p_{l}$ maintained in the oil liquid. At this stage, in the Equation (1) we neglected the pressure of saturated steam in the bubble and the change in bubble mass due to the diffuse processes of gas inside and outside the bubble. Moreover, assuming the process of expansion or contraction of the bubble is isothermal, we can write
(2)$$p_{g}=p_{g0}{\left(\frac{R_{0}}{R}\right)}^{3},$$
where $R_{0}$ is the initial bubble radius and $p_{g0}$ is the initial gas pressure in the bubble corresponding to some oil pressure $p_{l0}$ (say, $p_{l0}=100\;\mathrm{atm})$. Substituting (2) into (1) we get
(3)$$p_{l}R^{3}+2\sigma R^{2}-k=0,$$
Here, for convenience, we marked $k\;=\;p_{g0}R_{0}^{3}$. This algebraic equation determines the dependence of *R* with respect to values $p_{g0},\;p_{l}$ and $R_{0}$ which may vary in a wide range.

Figure 3 presents the numerical solution of Equation (3) as function of the oil pressure $p_{l}$ for different $p_{g0}$ and $R_{0}$. As it can be seen from the chart, with oil pressure decreasing from 100 to 10 atm, the equilibrium bubble radius is in the interval 0.5≤R≤2 μm. It means that we can take $R_{*}=1\;\mu m$ as a characteristic value in our following estimation.

In addition, in our case, we can neglect the oil degassing effect which manifests itself as gas bubbles floating to the surface of the liquid. Following the studies of [20,21], we shall look for the quasi-equilibrium state of gas bubbles under the Archimedes buoyancy force and resistance force. Restricting by small Reynolds numbers determined for the gas bubbles, the resistance force is given by the Stokes formula $F_{st}=-6\pi \eta R\left(\nu_{b}-U\right)$, where η is the dynamic liquid viscosity coefficient, $v_{b}$ is the velocity of the floating bubbles and U is the velocity of the oil flow. Then, equating the resistance force to Archimedes buoyancy force we can write the following balance equation:(4)$$6\pi \eta R(v_{b}-U)-mg=0,$$
where $m=\left(4/3\right)\pi \rho_{g}R^{3}$ is the mass of gas bubble with the density $\rho_{g}$ and g is the gravitational acceleration. From this relation, we obtain
(5)$$v_{b}-U=\frac{2}{9}\frac{\rho_{g}}{\rho_{l}}\frac{gR^{2}}{\nu },$$
here $\rho_{l}$ is the density of oil liquid and for clarity we use kinematic viscosity ν  related to the dynamic viscosity by the relation $\eta =\nu \rho_{l}$. In fact, the value $V_{r}=v_{b}-U$ is the relative velocity of floating bubble with respect to the oil flow. Although $\rho_{g}/\rho_{l}\ll 1$, to obtain the upper estimation of Equation (4) we put $\rho_{g}/\rho_{l}=1$ then for a gas bubble in oil ($\nu =10^{-4}\;m^{2}/s$ [2]), we get
$$V_{r}=2\times 10^{4}R^{2}.$$

For example, the relative velocity of floating bubble $V_{r}$ with $R_{*}=1\;\mu m$ according to this estimation is $V_{r}=2\times 10^{-8}\;m/s$. At the same time, the average oil velocity in the well is $U=10^{-1}\;m/s$ [12,22]. In fact, the gas bubbles are “frozen” into the oil flow and they drift with velocity U.

It should be kept in mind that some additional portion of gas can be brought into bubbles from the oil due to diffusion and convection [23,24]. As is known, the ratio between convective and molecular transport processes is represented by the Peclet number. In the case under consideration this value is defined as $Pe=V_{r}R/D$ where D is the diffusion coefficient of gas molecules in oil; as a characteristic estimate of this value, we take $D=10^{-6}\;{\mathrm{cm}}^{2}/s$ [2,24]. Then, for the considered range of parameters we have Pe≪1. It means that in the present case, we can expect some change in the amount of gas enclosed inside the bubble due to the balance between diffuse flows at the bubble boundary. That mechanism can be realized in quasi-equilibrium conditions; its existence and feasible effect shall be determined by the velocity, density and pressure of the flow.

## 4. Combined Hydrodynamic and Ultrasonic Activation

We may try to control the process of growing bubbles by changing the corresponding macroscopic flow parameters and using the external acoustic fields. Such a combined action can be carried out in the scheme shown in Figure 4 where the narrowing area alternates with the expansion area of hydrodynamic tract. In this schematic, we expect to have the superimposition of two processes: a fast external acoustic impact and a slow change of parameters of the hydrodynamics flow which, moreover, propels the process of gas diffusion from the surrounding oil medium into the gas bubbles. It is worth noting that the intensity of such process shall be defined by the relationship between $d_{1}$ and $d_{2}$, because of the geometry for the hydrodynamic tract changes. This immediately follows from the Bernoulli’s law:$$\frac{\rho_{1}v_{1}^{2}}{2}+p_{1}=\frac{\rho_{2}v_{2}^{2}}{2}+p_{2},$$
where $\rho_{i},p_{i}$ are the density and pressure of the flow at the corresponding part of the scheme (i=1,2 for the part with diameter $d_{1}$ and $d_{2}$, respectively). Here, we have considered the case where $\rho_{1}\approx \rho_{2}$, since the $d_{1}<d_{2}$, we get $v_{1}<v_{2}$. Therefore, the pressures obey the inequality $p_{1}>p_{2}$. In the described situation, due to the increase in pressure when the flow moves from the second region of scheme to the first, the energy accumulated by the bubbles in the second region is released in the first. The released energy converts into the kinetic energy of the particles surrounding the bubbles, which can provoke the break of molecular bonds in the surrounding fluid.

Such scheme may be located at the exit from the formation to the well (see Figure 4). At the region of acoustic waveguide, there may be an additional oil pressure drop as opposed to the bottom of the well. This drop can be brought about from the accelerated growth of bubbles coming from the reservoir into the well. Moreover, this may propel the growth of some microscopic voids under-filling them with the gas contained in oil medium (see, for example, [21,23,25]).

As a result, a two-phase medium may be formed containing a huge number of bubbles, i.e., the density of medium in the area of the acoustic waveguide may decrease significantly, which can facilitate the oil lift from great depths. In fact, here, we expect to get an internal airlift when gas bubbles are formed due to internal diffusion processes occurring in the volume of oil medium under the ultrasonic action.

Now, we move on with the discussion of the physical and technical conditions necessary for the implementation of such a script. In the case under consideration, instead of the Laplace Equation (2), we must write
(6)$$p_{g}=p_{l}+\frac{2\sigma }{R}-p_{a},$$
where
(7)$$p_{a}=p_{a0}sin\left(\omega t\right),$$
where $p_{a0}$ and ω are the amplitude and frequency of acoustic pressure to be determined.

In order to obtain the rough estimation of bubble dynamics, we study the case when gas flow from the bubble is negligible and the change of gas density $n_{g}$ in the bubble can occur due to the change in the bubble volume as well as the gas inflow from the surrounding liquid. Then, the equation of gas balance for one bubble reads
(8)$$\frac{d}{dt}\left(n_{g}R^{3}\right)=3R^{2}j_{g},$$
where the gas inflow $j_{g}$ from oil liquid under Pe≪1 is determined by the relation [23]:(9)$$j_{g}=\frac{Dn_{\infty }}{R}\left(1+\frac{R}{\sqrt{\pi Dt}}\right),$$
where $n_{\infty }$ is the gas density in oil far from the bubble. For time $t\gg R^{2}/D$, we can neglect the second term in Equation (9). Hence, we may use
(10)$$j_{g}=\frac{Dn_{\infty }}{R}.$$

Restricting our consideration by the isothermal case and using the state equation for ideal gas, $p_{g}=Tn_{g}$, where gas temperature T=const, from (6) we find
(11)$$n_{g}=\frac{p_{l}}{kT}+\frac{2\sigma }{kTR}-\frac{p_{a}}{kT}.$$

Substituting (11) and (10) into (8), allowing for $dp_{l}/dt\approx 0$ we get
(12)$$\frac{dR}{dt}=\frac{3kTDn_{\infty }+\omega p_{a0}R^{2}cos\left(\omega t\right)}{3p_{l}R+4\sigma -3p_{a0}Rsin\left(\omega t\right)}.$$

This equation must be supplemented by the initial condition $R\left(t=0\right)=R_{0}$. By using dimensionless variables
$$\tau =\omega t,\;\;r=\frac{R}{R_{0}},$$
it is convenient to restate Equation (12) as
(13)$$\frac{dr}{d\tau }=\frac{3\xi F_{0}+\kappa r^{2}cos\left(\tau \right)}{3r\left[1-\kappa sin\left(\tau \right)\right]+\zeta },$$
where $\xi =kTn_{\infty }/p_{l}$ is the gas content in oil, $F_{0}=D/\omega R_{0}^{2}$ is the diffuse Fourier number, $\kappa =p_{a0}/p_{l}$ is the ratio of the acoustic pressure amplitude to the oil pressure and $\zeta =4\sigma /R_{0}p_{l}$ is the surface-to-fluid pressure ratio. Depending on these dimensionless parameters, Equation (13) admits a rich variety of solutions. Therefore, we shall select the range of basic parameters, where there is an increase in bubble radius with allowance for the existing technical capabilities.

So, the gas content in oil can vary from 10 to 50%, which gives 0.1≤ξ≤0.5. The magnitude of $p_{a0}$ is set by the power of acoustic generator W which can vary in the range $10^{4}–10^{5}\;W/m^{2}$ (see, for example, [26,27] and references therein). On the other hand, this power goes on the excitation of an acoustic wave which transfers the energy flow. As an estimation of such an energy flow, we take the relation for linear acoustic waves propagating in a homogeneous medium:$$I=\frac{p_{a0}^{2}}{\rho_{l0}c_{0}},$$
where $\rho_{l0}$ is the density of oil medium and $c_{0}$ is the velocity of sound in unperturbed medium. It should be noted that here we have neglected the dependence of the sound velocity on the content of bubbles in the oil; this shall slightly increase the sound velocity and decrease the amplitude $p_{a0}$. Then, proceeding from assumption I~W we get
(14)$$p_{a0}=\sqrt{c_{0}\rho_{l0}W}.$$

As typical values for oil in natural conditions, we take $c_{0}=1225\;m/s$ and $\rho_{l0}=900\;\mathrm{kg}/m^{3}$ [2,3]. Then, using (14), we find that amplitude $p_{a0}$ changes in interval $1\leq p_{a0}\leq 3\;\mathrm{atm}$ for the chosen range of W. It means that in the bottom hole, the parameter κ can vary in the range of 0.1≤κ≤0.3. Since ζ>0, the denominator of Equation (13) never vanishes, namely, the singular dynamics for the model is excluded.

Underestimating admissible frequencies, we shall proceed from point that the diffusion process prevails over acoustic impact. Such a script comes about when the characteristic diffusion time $\tau_{d}=R_{0}^{2}/D$ is less than the characteristic time of acoustic impact $\tau_{a}=\omega^{-1},$ i.e., $F_{0}\geq 1$. It implies that
(15)$$\omega \leq \frac{D}{R_{0}^{2}}.$$

On the other hand, if we wish to obtain an effect on the entire volume of liquid in the region of the acoustic resonator, we should require the acoustic wave length must be commensurate with the transverse dimension of the waveguide: $2\pi c_{0}\omega \leq d_{1}$. From this condition, we get
(16)$$\omega \geq \frac{2\pi c_{0}}{d_{1}}.$$

In order to avoid the consideration of resonant acoustic phenomena (e.g., the acoustic radiation force) along with condition (15), we have to require
(17)$$\omega \leq \omega_{0},$$
where $\omega_{0}=\sqrt{3p_{l}/\left(\rho_{l0}R_{0}^{2}\right)}$ is the monopole resonance frequency of bubble with characteristic radius $R_{0}$. Thus, the relations (15)–(17) determine the allowable technical parameters of the system worked out.

Now, let us pass over to the numerical study of the model Equation (13) with the conditions described above. The solution of Equation (13) for variable generator frequency in provided constraints is shown at the Figure 5. As we can see the amplitude of radius’ oscillations is higher when the generator frequency is lower. In addition, the velocity of radius rising is higher when the frequency is lower. For variable oil flow pressure, the results are shown at the Figure 6. One can see here that the amplitude is higher when the pressure is lower, but the differences are barely noticeable.

As is seen from Figure 5 and Figure 6, presented results indicate a strong need to select the generator frequency, while the resulting effect on the flow is weakly dependent on its pressure. In addition, the excitation of acoustic oscillations in the flow of oil–liquid, which in a given way is either accelerated or slowed down, can provide a local effect on the structure of the studied medium.

## 5. Features of the Formation of Thermal Fields in Dispersed Systems and Simple Liquids

Following the studies of [9,12,22], we consider the manifestation of these features in an example gas bubble formation process in oil (hereinafter, this process is called oil degassing) where the temperature changes due to the heat of oil degassing. In this case, the temperature change should depend on the oil saturation of the formation. Herewith, the presence of water in reservoir due to the lower solubility of hydrocarbon gases should lead to ever decreasing heat effect of oil degassing on the generated temperature field.

Hence, the information about the features of the temperature field formation may be associated with the oil saturation of formation. In other words, this feature makes it possible to assess porous medium saturation with oil using the effect of acoustic field on such a medium. In order to describe these processes, we use the simple model reported in [9,12] for the thermal field formation in an oil-saturated porous medium under acoustic impact, which may initiate the formation of bubbles. 

Let us briefly dwell on the physical points underlying this model. We shall consider the porous medium saturated with oil, where a local decrease in pressure occurs. This decrease in pressure leads to carbonated oil filtration while flowing from the formation to the well. Under the influence of these factors in the region ($r\leq r_{b}$), where $r_{b}$ is the characteristic radius of the degassing region, the gas bubbles form, that leads to a change in temperature for the selected region due to the heat of degassing, adiabatic and Joule-Thomson effects. To reveal the influence of such heat sources in the first approximation we neglect the heat losses to the surrounding rocks. The convective heat transfer is assumed to prevail over thermal conductivity along the path of fluid movement, i.e., one can neglect the thermal transfer effects. In addition, we give up the consideration of oil evaporation. The degassing process is assumed to be in equilibrium, obeying Henry’s law. This allows us to distinguish three phases in the system under the consideration: the skeleton of porous medium, oil and gas phases. 

Considering these points, let us pass over to the numerical analysis of the thermal field formations in an oil layer. The corresponding hydrodynamic and thermophysical parameters are taken from the work [22,28] and we used initial and boundary conditions similar to that in Refs. [9,12,28,29]. 

Successively, we should take a look at what happens when the system contains: (1) oil and gas, (2) water and gas and (3) water. The numerical results for these cases are shown in Figure 7. Herewith, it should be noted that in cases (1) and (2) the gas properties are the same. As it is seen from these graphs, the greatest temperature decrease is observed in the oil-saturated reservoir with dissolved gas. There is a rapid decrease in oil temperature here, as the oil cooling brought about the degassing process prevails over heating due to the Joule–Thomson effect. In the case of the water-saturated reservoir, after a slight decrease in temperature, there is a positive temperature anomaly (relative to the geothermal one at a given depth) when the increase of temperature comes about with respect to time. 

This behavior is caused by the small amount of gas released from the water. In the present case, a slight cooling of water comes about because of adiabatic effect and gas evolution with decreasing pressure. However, this effect is somewhat less than heating due to the Joule–Thomson effect for water. As a result, we have a situation when the second effect always prevails the first one. From the graphs in Figure 7, the basic result is that depending on the composition of the liquid the sign of the temperature derivative changes and this feature can be used to diagnose the oil saturation of the reservoir. This issue will be discussed in more detail in the next section. 

## 6. Comparison of Model and Field Data

Now, we consider the thermograms obtained during oil production in a well to confirm the connection between the oil saturation of the medium and its temperature changes. The picture of these observations for the well No 1 is given in Figure 8. This well contains two permeable formations located at a depth of 2411.2–2413.2 m and 2415–2422 m, respectively. The short-term inflow of fluid from the reservoirs was carried out using a compressor unit. It is worth noting that these graphs were obtained in [28] and are used to compare our analysis and field data.

The analysis of the present thermograms shows that the throttling heating is observed when there is fluid inflow from the lower perforated formation (curve 2 in Figure 8). After the compressor is turned off, the pressure in the well decreases below the bubble-point pressure for oil. At the same time, one can see the decrease in temperature (curve 3 in Figure 8) against the lower layer. The shape of the temperature dependence being above the inflow zone indicates the predominant inflow of gas phase into the wellbore. The curve 4 in Figure 8 corresponds to some pressure increase, but it is still less than the bubble point pressure; in this case, there appears an inflow of the gas-oil mixture [30]. 

One can observe that the curve 5 in Figure 8 relates to the conditions when the well pressure is higher than the bubble-point pressure, and the fluid inflow is observed from the formation. Such a heating anomaly may be considered as a sign that the lower layer is oil saturated. Herewith, a decrease in pressure without gas release from oil would only lead to an increase in the positive temperature anomaly. Essentially, comparing the signs and values of temperature derivatives for oil flow with gas bubbles with respect to pure water flow and water–oil liquid, a practical conclusion on the water content in the well can be made. Proceeding from the described idea, it is possible to compare the magnitude and direction of the temperature gradient in the well with the oil concentration gradient in it.

Namely, such vivid behavior is observed in Figure 9 which illustrates the change in temperature anomaly after the waterflooding in relation to the anomaly in the flow of pure oil. In this well, three layers are perforated. The first studies of the temperature distribution were carried out before waterflooding the reservoirs when pure oil is produced from the well. As is seen from dependence T1 in Figure 9, the temperature decrease comes in all three working layers when the wellbore pressure falls below the bubble-point pressure. Curve T2 in Figure 9 represents the case where the pressure increases when the well begins to shut. This process is accompanied by the temperature increase of medium due to the throttling heating. From the modeling above, it follows that such behavior of the thermograms is caused by the high oil saturation of layers. Repeated studies performed three years after well startup show changes in the nature of the temperature distribution over depth (see curves T3 and T4 in Figure 9). It is worth noting that, opposite to the lower layer, during the operation of the well and after its shutdown, the temperature anomalies are positive; hence, there is a throttling heating of the liquid. Whereas the upper layers are characterized by the negative temperature anomaly (the temperature decrease), as in the previous case, this means that the lower layer has been watered—water comes out from it. 

Thus, the analysis of the given production data indicates that it is possible to apply the results of modeling presented in Section 5 for seeking oil fields. Furthermore, the usage of acoustic fields could induce both temperature effects in wells and structural changes in oil flows, simplifying its production [31].

## 7. Concluding Remarks

In the present paper, we have discussed the structural features of dynamics for high-molecular systems (e.g., the polymer fluid or liquid dispersed system) saturated with gas under changing the flow pressure. As a typical sample of these systems, we have taken the oil; that is defined by the existing and possible applications of this medium. We have considered the physical case when the formation of gas bubbles comes about in the natural voids of the dispersed system (in the fact, such voids are the nuclei of bubbles) under governing the pressure of medium. The pressure change can be caused by two factors: changing the geometry of the tract in which the medium moves and the impact of external acoustic fields on the medium. Here, we have considered the model where both factors have been considered. The use of this model makes it possible to study a bubble dynamic with respect to the flow pressure and the outgassing from the medium. Based on the model, the bubble radius oscillations were simulated depending on the parameters of the acoustic field and the pressure in the flow. 

We tried to connect the results about the bubbles dynamics to analyze the temperature changes in a medium on pressure. In the worked-out case, we found the relationship between temperature gradients in the medium depending on the change in pressure for various types of saturation. There is a strong temperature dependence on composition of a two-phase medium, especially the water content in liquid. As a result, the opposite dependence of temperature gradients has determined during the degassing of an oil-saturated medium and a water–gas medium. In a water-saturated medium, the temperature increases during degassing, primarily due to throttle heating; in an oil-saturated medium, we observe the opposite effect: the temperature decreases during degassing. Proceeding from this point, the dependence between the temperature changes and the changes in oil saturation of medium has been established. 

These results should be helpful in diagnosing oil fields [32]. As is seen from the analysis of dependences presented in Figure 7, Figure 8 and Figure 9, to conduct this analysis, one should have the data on well thermograms obtained during oil production. However, in order to fully understand the range of the worked-out model, it will be necessary to considerably extend the analysis of external acoustics impacts and internal processes of the structure formation for oil dispersed medium for real natural conditions.

## Figures and Tables

**Figure 1 polymers-14-01497-f001:**
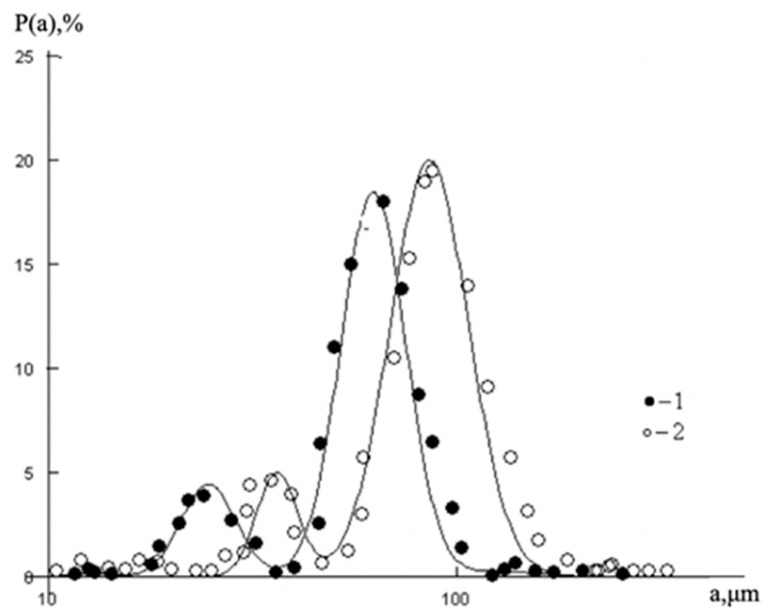
Emulsion particle size distribution within mixing after time 1–10 min; 2–20 min [16].

**Figure 2 polymers-14-01497-f002:**
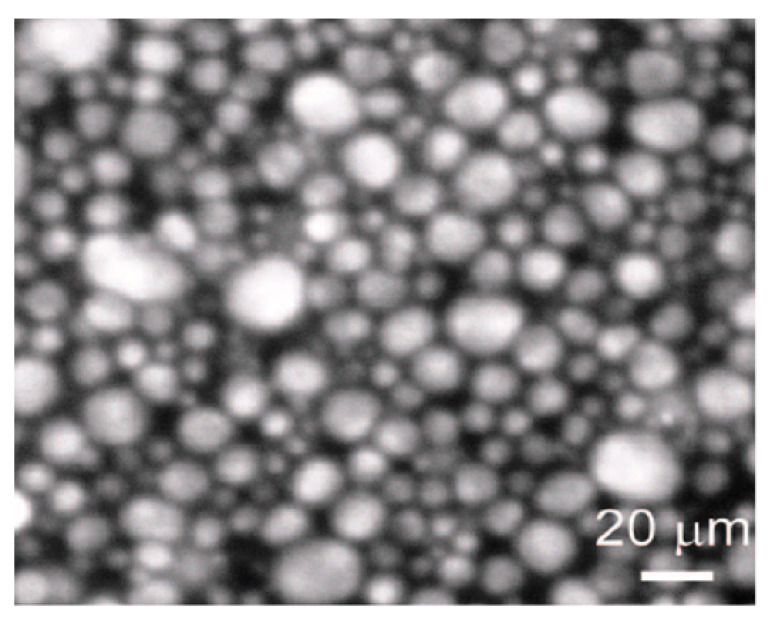
Photo for microstructure of water–oil system stabilized by particles of alum. oxide [13].

**Figure 3 polymers-14-01497-f003:**
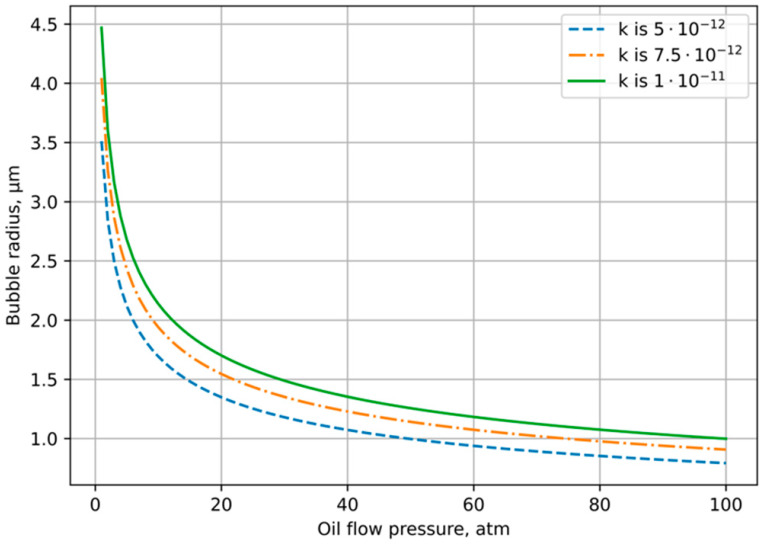
The graph of the radius dependence on the fluid pressure with different k.

**Figure 4 polymers-14-01497-f004:**
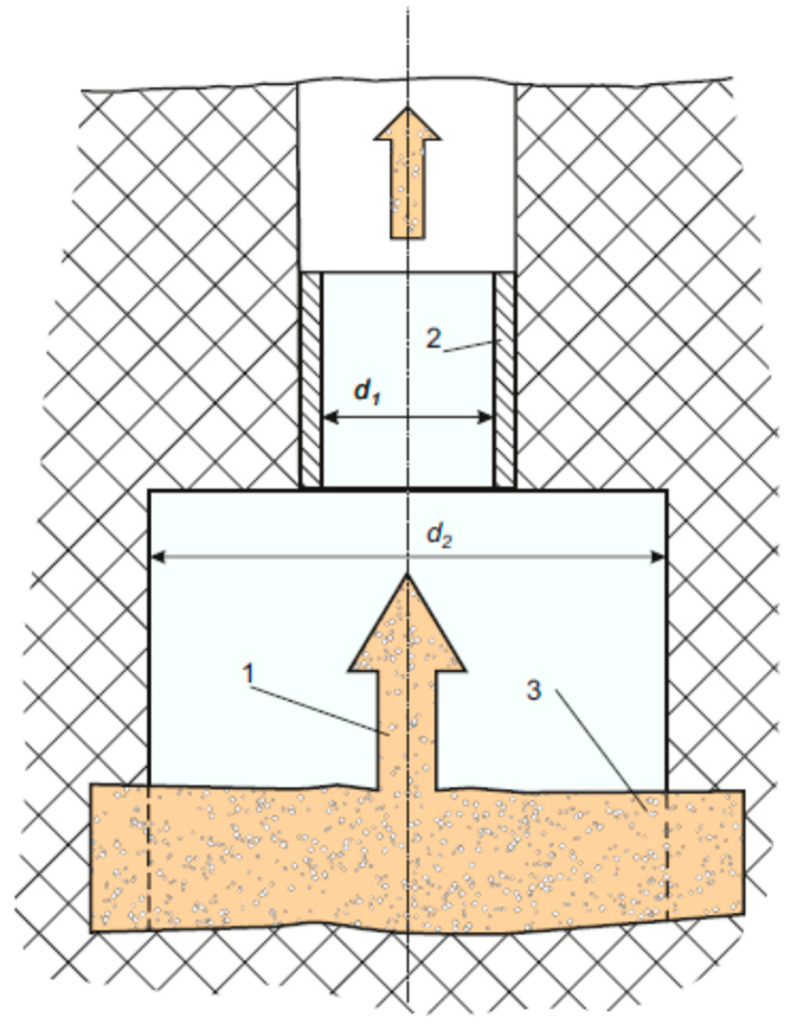
Schematic of hydrodynamic tract: 1—oil flow, 2—acoustic emitter, 3—oil-saturated reservoir.

**Figure 5 polymers-14-01497-f005:**
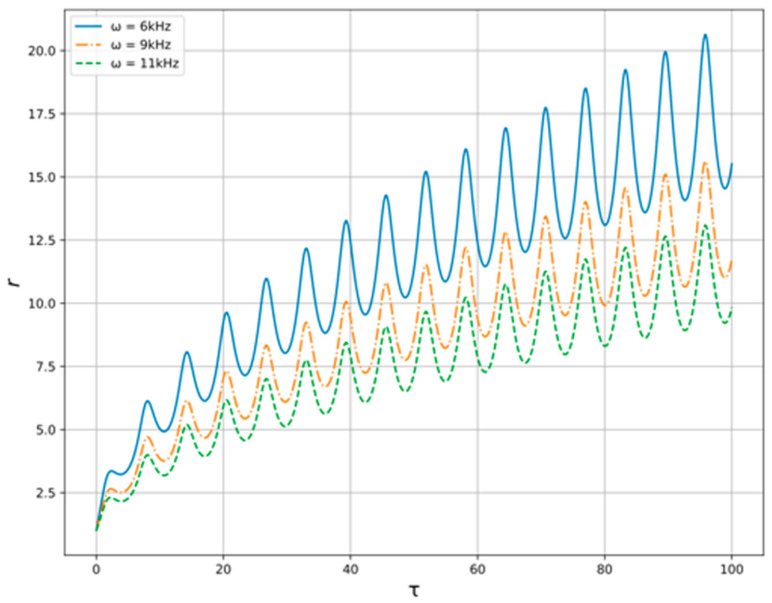
Bubble radius oscillations for the variable generator frequencies ω.

**Figure 6 polymers-14-01497-f006:**
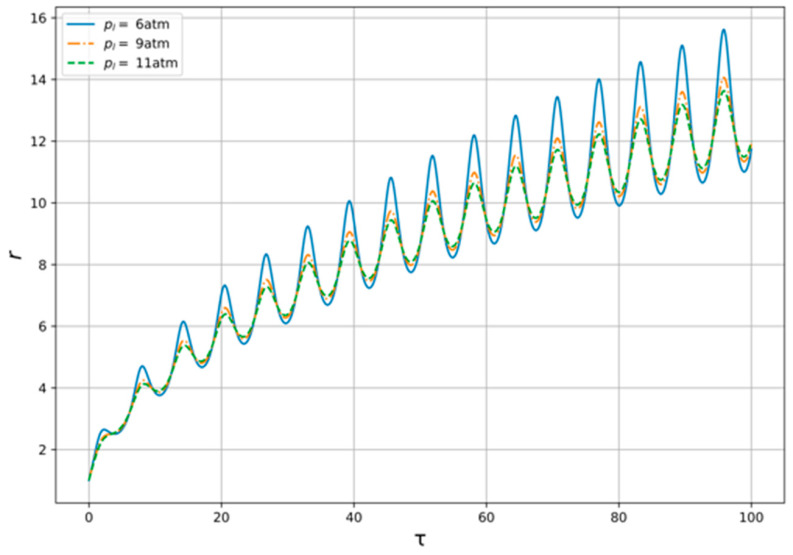
Bubble radius oscillations for the variable $p_{l}$.

**Figure 7 polymers-14-01497-f007:**
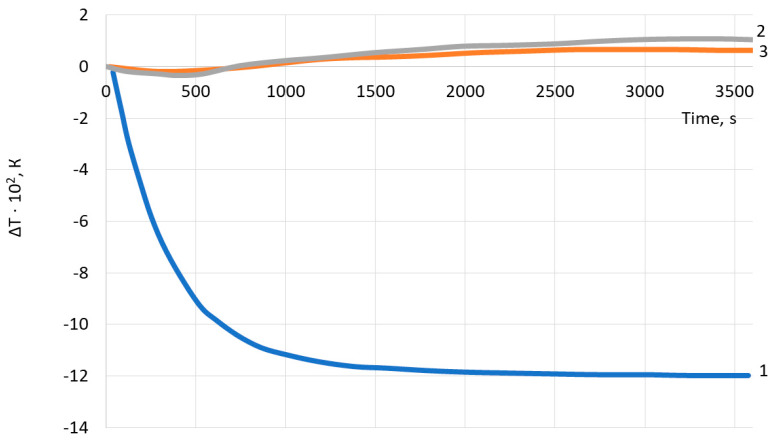
Dependence of the sandface temperature change vs. time for different compound: 1—oil + gas, 2—water + gas, 3—water.

**Figure 8 polymers-14-01497-f008:**
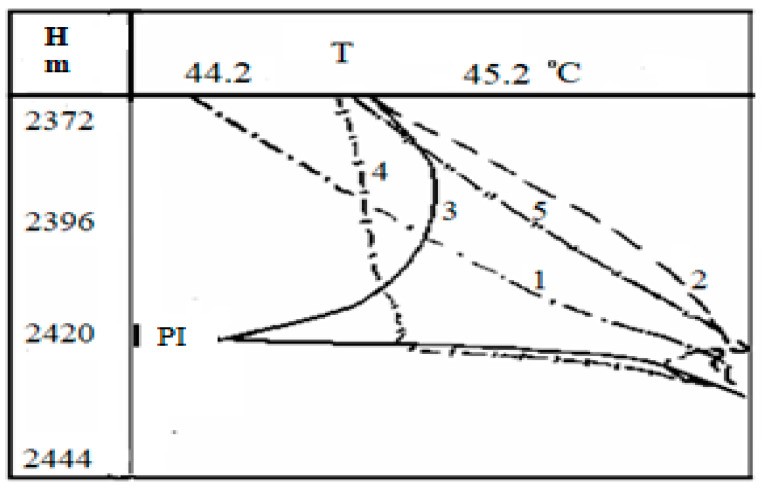
The results of well test. 1—background, before the compressor is switched on; 2—when the compressor is running; 3—immediately after shutting down the compressor and reducing the pressure in the well; 4 and 5, 1 and 2 h after the compressor is turned off (PI is the perforation interval) [28].

**Figure 9 polymers-14-01497-f009:**
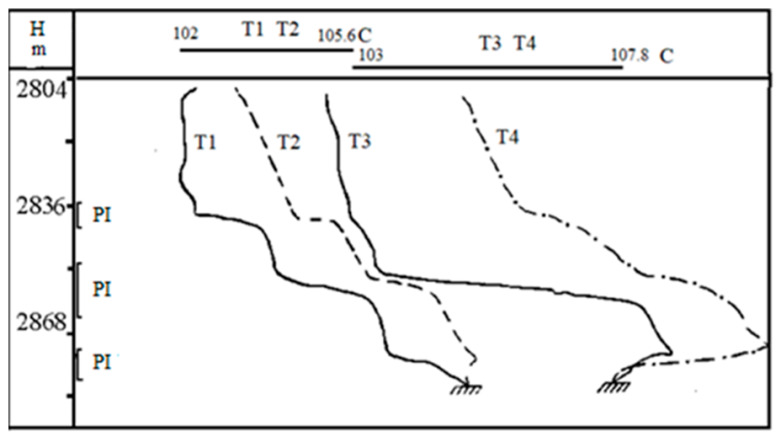
The results of well test. Here, curves T1 and T3 relate to a working well, curves T2 and T4 relate to a shut-in well (PI is the perforation interval) [28].

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
