# Peer review of "The Degassing Processes for Oil Media in Acoustic Fields and Their Applications"

_polymers, 2022, doi:10.3390/polym14081497_

Round 1
Reviewer 1 Report
The authors presented detailed theoretical analysis of degassing processes for oil media Not much major comments for the manuscript, but with some minor comments listed below:
- Abstract: Can elaborate the results or mechnanism with more details, preferably with quantification results.
- Make sure that copyright permission is obtained when reusing figures from others’ works or publications
- Eq(1): Is it ‘p_L’ or ‘p_1’. Please be consistent with the symbols presented and description in the text.
- Seems like an error for the symbols used in this phrase: “ ? = 10−2 ÷ 10−1 ?/s”
- Figures 8-9, can the authors explain which curve is the results from the authors’ own analysis? It seems to me the curves are mainly based on previous works
- In conclusion, perhaps can also elaborate more on the results of temperature dependence on composition of water content of two-phase medium.
Author Response
Thank you for your review! More comments in the pdf-file.

Reviewer 2 Report
This paper presents a review about the degassing processes for oil media in acoustic fields and their applications. Overall, the paper's topic is interesting for the reader; however, the following issues need to be clarified
1- The language level must be improved.
2-Short paragraphs are not desired, please combine the most of them with a harmony of relevance.
3- Introduction section is very short, please first give some information about the scope and the topic. At the end of the introduction section please emphasize the novelty and/or necessity of this review.
After finalizing the editing process with the points above, I will read again the manuscript for further revision.
Author Response
Thank you for review! More comments in the cover letter.

Round 2
Reviewer 2 Report
The paper can be accepted in the present form.
This manuscript is a resubmission of an earlier submission. The following is a list of the peer review reports and author responses from that submission.
Round 1
Reviewer 1 Report
The novelty level of this manuscript is very low. There are many articles in this field. The study does not contribute substantially to the Polymers and may not be widely read by the polymers community. The manuscript also requires many more tests to substantiate the authors' claims and it offers no critical information and no new slant on the topic. The descriptions of the results are very superficial and need to be re-examined. In addition, the references are very old and the literature review is poor. So, the quality of this manuscript does not correspond to the journal of Polymers.
Reviewer 2 Report
The revised manuscript is significantly improved by the authors.
Reviewer 3 Report
All questions have been addressed clearly. The manuscript is recommended for publication in Polymers.Reviewer 4 Report
Due to the declining resources of fossil fuels (oil), methods are being sought to maximize the use of this raw material. The processing of crude oil is economically important. The authors of the reviewed work undertook to explain the oil degassing process using the acoustic field and the potential practical applications of this solution. According to my opinion, the work is a review of the literature and previously published works of the authors (mainly in the mother language - russian). The way of writing the reviewed work is rare, written personally (we my try ...), not very common. Receipt of work is limited. Requires editorial proofreading (line 43 dwell, line 420 especially especially and others). Drawings (e.g. 1) of poor quality.